# Improved detection of *SBDS* gene mutation by a new method of next-generation sequencing analysis based on the Chinese mutation spectrum

**Dong Wu[1], Li Zhang[2], Yuzhen Qiang[2], Kaiyu Wang[2]***

**1** Department of Obstetrics and Gynecology, 900 Hospital of the Joint Logistics Team or Dongfang Hospital, Fuzhou, Fujian, People's Republic of China, **2** Fulgent (Fujian) Technologies, Fuzhou, Fujian, People's Republic of China

* wangkaiyu@fulgent.com.cn

**Data Availability Statement:** All relevant data are within the paper and its Supporting Information files.

## Abstract

Next-generation sequencing (NGS) is a useful molecular diagnostic tool for genetic diseases. However, due to the presence of highly homologous pseudogenes, it is challenging to use short-read NGS for analyzing mutations of the Shwachman-Bodian-Diamond syndrome (*SBDS*) gene. The *SBDS* mutation spectrum was analyzed in the Chinese population, which revealed that *SBDS* variants were primarily from sequence exchange between *SBDS* and its pseudogene at the base-pair level, predominantly in the coding region and splice junction of exon two. The c.258+2T>C and c.185_184TA>GT variants were the two most common pathogenic *SBDS* variants in the Chinese population, resulting in a total carrier frequency of 1.19%. When analyzing pathogenic variants in the *SBDS* gene from the NGS data, the misalignment was identified as a common issue, and there were different probabilities of misalignment for different pathogenic variants. Here, we present a novel mathematical method for identifying pathogenic variants in the *SBDS* gene from the NGS data, which utilizes read-depth of the paralogous sequence variant (PSV) loci of *SBDS* and its pseudogene. Combined with PCR and STR orthogonal experiments, *SBDS* gene mutation analysis results were improved in 40% of clinical samples, and various types of mutations such as homozygous, compound heterozygous, and uniparental diploid were explored. The findings effectively reduce the impact of misalignment in NGS-based *SBDS* mutation analysis and are helpful for the clinical diagnosis of *SBDS*-related diseases, the research into population variation, and the carrier screening.

## 1. Introduction

High-throughput, high-sensitivity, and low-cost next-generation sequencing (NGS) have made tremendous progress in clinical molecular testing. Analysis with NGS accurately obtains information on various types of mutation by repeatedly reading the sequence of each base and

**Funding:** The author(s) received no specific funding for this work.

**Competing interests:** The authors have declared that no competing interests exist.

searching for the optimum matching coordinates in the reference genome [1]. However, when a gene region has one or more pseudogenes with high sequence homology, the short NGS reads cannot be aligned accurately to the target locus, increasing the likelihood of false positives or false negatives [2]. Misalignment is more likely to occur, especially when there is a conversion between gene and its pseudogene, presented by the exchange of one or multiple paralogous sequence variants (PSVs) [3]. Therefore, the interference caused by pseudogenes as well as their conversion is one of the key problems with the clinical application of NGS.

The protein encoded by the Shwachman-Bodian-Diamond syndrome (*SBDS*) gene is involved in ribosomal RNA processing. Since its first publication in 2003, pathogenic variants in the *SBDS* gene have been detected in more than 90% of people with Shwachman-Diamond syndrome (SDS) [4]. Also known as Shwachman-Bodian-Diamond syndrome, SDS is an autosomal recessive genetic disease involving multiple systems. Its clinical manifestations include exocrine pancreatic dysfunction (chronic diarrhea), skeletal metaphysis development disorder, and different degrees of bone marrow dysfunction accompanied by hypocytosis [5,6]. About one-third of patients suffer from myelodysplastic syndrome and acute myeloid leukemia [7]. Other clinical symptoms include short stature, liver dysfunction, susceptibility to infection, nephrocalcinosis, myocardial necrosis, and neonatal respiratory distress [8,9]. Dozens of *SBDS* pathogenic variants have been reported to cause SDS, the most common of which include c.185_184TA>GT, c.258+2T>C, and c.[185_184TA>GT;258+2T>C]. A study used targeted analysis for the three common *SBDS* pathogenic variants and detected at least one pathogenic variant in 90% of affected individuals and both pathogenic variants in approximately 62% of affected individuals who had SDS along with *SBDS* [4]. Nonetheless, multi-gene testing using NGS is still recommended in the molecular diagnosis of suspected cases of SDS that can be caused by pathogenic variants in other genes besides *SBDS*. Additionally, the phenotypic spectrum of SDS is broad, and multi-gene testing might be the best option for an individual with atypical phenotypic features [10].

The *SBDS* gene has a highly homologous pseudogene, *SBDSP1*, which is located ~ 5.8Mb downstream of the *SBDS* gene at chromosome 7q11. The five exon coding sequences of *SBDS* have 96.8% (95%–98.2%) homology with *SBDSP1*. Since the common *SBDS* pathogenic variants c.185_184TA>GT and c.258+2T>C are at the functional PSV loci, it has been suggested that recombination and gene conversion might occur between *SBDS* and *SBDSP1* [11]. Based on the above factors, researchers have recognized the complexity and challenges of NGS analysis of the *SBDS* gene, and therefore suggest using PCR, RT-PCR or specific bioinformatics tools to increase the sensitivity and specificity of detecting pathogenic variants in *SBDS* [12–14].

This study focused on analyzing the *SBDS* mutation spectrum through large-scale whole-exome sequencing (WES) within the Chinese population. After determining the misalignment issues in the NGS data analysis, we tried to overcome these problems by developing a novel mathematical method using NGS data to analyze the pathogenic variants of the *SBDS* gene.

## 2. Materials and methods

### DNA samples

To study the *SBDS* mutation spectrum of the Chinese population and the misalignment of NGS data, we selected 20,542 de-identified samples that were sent to Fulgent (Fujian) Technologies Co., Ltd.(Short for "Fujian Fulgent") for WES by NGS from October, 2019 to March, 2021. Authors could not access to information that could identify individual participants during or after data collection after application. All research participants were from the Chinese population, and their EDTA anti-coagulated venous whole blood was obtained after signing

their informed consent. Samples were divided into non-SDS and suspected SDS groups according to whether the individual's clinical diagnosis included SDS phenotypes, such as chronic diarrhea, cytopenia, or liver dysfunction. The non-SDS group included 20,395 cases, in which none of the above phenotypes were detected. The suspected SDS group contained any two of the above phenotypes and included a total of 147 cases. Since most of the samples sent to Fujian Fulgent were from suspected cases of genetic diseases, the grouping criteria did not include common phenotypes of genetic diseases, such as growth failure and bone abnormalities, but included liver dysfunction, which is reportedly more common in people from China with SDS. According to the manufacturer's instructions, the genomic DNA was extracted from whole blood using commercially available DNA isolation kits. This study received approval from the ethics committee of No. 900 Hospital of the Joint Logistics Team.

## Whole-exome sequencing

Genomic DNA samples were analyzed using the Illumina HiSeq or NovaSeq platforms (Illumina, USA) with PE150 (paired-end sequencing, 150-bp reads). Briefly, DNA libraries were prepared using the IDT xGen Exome Research Panel v2 exome library (71Mb target region). Samples from the same capture pool were grouped together. The raw sequence data were aligned to the human reference genome GRCh37/hg19 using Novoalign v3.09.00 software (Novocraft Technologies, Malaysia). SNPs and short indels were called using SAMtools v1.3.1 and VarScan software v2.3.9. Variants were annotated using Alamut-Batch standalone v1.9 software (Interactive Biosoftware, France). The average read-depth of each exon of *SBDS* and the corresponding region of *SBDSP1*, as well as the read-depth of their PSVs loci, were extracted with BEDTools coverage (v.2.30.0) using the options "-d -a". Variants were curated manually according to the American College of Medical Genetics (ACMG) scoring system [15].

## Sanger sequencing

The primer sequences provided by Woloszynek were employed to amplify the five exons, splice sites, and intron sequences flanking the exons of *SBDS* [16]. Amplicons were generated using 5 ng of genomic DNA using Ex Taq polymerase (Takara, China). The PCR conditions for this amplification were as follows: 95˚C for 5 min, 35× (95˚C for 30 s and 60˚C for 15 min), 72˚C for 5 min. Products were verified by agarose gel electrophoresis and then sequenced using the sequencing primers on a 3730 genetic analyzer (Applied Biosystems, USA).

## STR typing

In the parentage test, DNA samples were detected with Microreader™ 21 ID System (Suzhou Microread Genetics, Suzhou, China), according to the standard procedures of the manufacturer.

## 3. Results and discussions

Previous studies have speculated that *SBDS* pathogenic variants may be converted from *SBDSP1* because these pathogenic variants often occur at the functional PSV loci of *SBDS* and sometimes include other cis variants at PSV loci [11]. Therefore, research has sought to determine if the formation of *SBDS* pathogenic variants occurs through gene conversion and how frequently this conversion takes place. This information is very important for formulating NGS data analysis strategies for *SBDS* genes and validating the accuracy of the results [2]. This study aimed to acquire this information through large-scale WES NGS data analysis. In

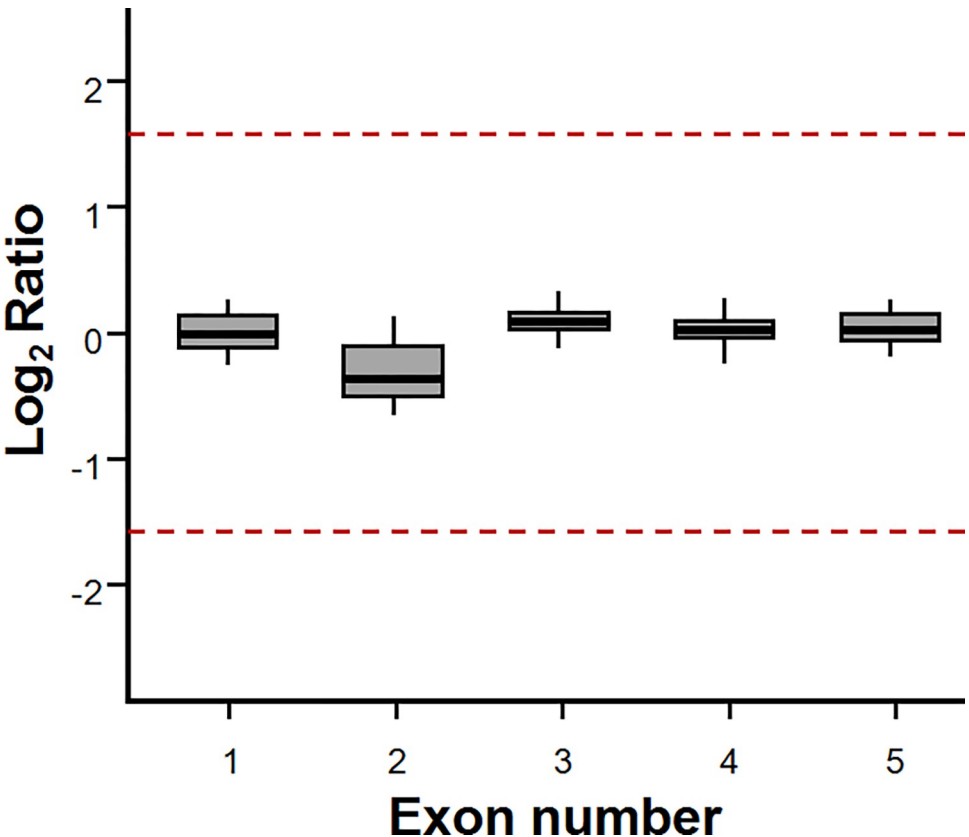

**Fig 1. The misalignment analysis of *SBDS* to detect gene conversion.** The red lines are the expected the read-depth ratio values when the conversion between *SBDS* and *SBDSP1* occurs. The median read-depth ratio of *SBDS* to *SBDSP1* among 2000 random samples in non-SDS group. The error bars represent the standard error of the median read-depth ratio.

general, the average read-depth ratio of a gene to its pseudogene will significantly shift in genes or regions where gene conversion is more frequent, such as exons 13–15 of the *PMS2* gene [17]. The average read-depths of each exon of *SBDS* and the corresponding region of *SBDSP1* were extracted to estimate the frequency of conversion between *SBDS* and *SBDSP1*. Subsequently, the read-depth ratios of *SBDS* to *SBDSP1* were calculated from 2000 WES samples, which were randomly selected from the Chinese population in the non-SDS group. Theoretically, when *SBDS* is converted to *SBDSP1*, the read-depth of *SBDS* becomes lower than that of the corresponding *SBDSP1* region, thereby decreasing in the ratio of *SBDS* to *SBDSP1*. When the opposite conversion occurs, the ratio increases. In this study, the average read depth of each exon of *SBDS* was not significantly different from that of the corresponding region of *SBDSP1* (Fig 1), indicating that the *SBDS* NGS data can usually be accurately aligned, while gene conversion of *SBDS* and *SBDSP1* in exon-level or in a larger range is not common.

Further analysis aimed to determine if the interlocus conversion between the *SBDS* and *SBDSP1* genes occurred at the sub-exon level [18,19]. Since this conversion was manifested as the exchange of PSVs in NGS analysis, we tried to obtain the *SBDS* mutation spectrum from WES data of 20,395 samples from the Chinese population. A total of 64 different *SBDS* variants were discovered, of which seven were detected more than ten times (S1 Table). Among them, c.141C>T, c.258+2T>C, and c.201A>G were the top three most frequently detected variants, with carrier frequencies of 4.16%, 1.01%, and 0.61%, respectively, and all were at PSV loci of

*SBDS*. The total carrier frequency of ten variants at PSV loci of *SBDS* was 6.18%, which was significantly higher than fifty-four others at non-PSV position (1.23%). In addition, through analyzing the NGS data of *SBDSP1*, the frequency of these common variants in *SBDS* were found to vary in accordance with variants in the corresponding *SBDSP1* loci ($P < 0.05$), but did not result from the reciprocal sequence exchanges because there was no significant correlation between their occurrence ($P > 0.05$) (S2 Table). This observation indicated that the variants of *SBDS* mainly originated from the gene-conversion event, the frequency of which was very low in the Chinese population. Furthermore, we observed that most variants at PSV loci of *SBDS* were detected in isolation, and only 35 samples were found to have in-cis variants at adjacent PSV loci, all of which were in exon two and its flanking sequence of the *SBDS* gene. According to the distance between PSVs in the *SBDS* coding region, it can be inferred that sequence conversions between *SBDS* and *SBDSP1* mainly occurred at the base pairs level, and longer conversions at the sub-exon level only occur in the Exon two region.

In order to analyze the pathogenic variant spectrum of *SBDS* in the Chinese population and provide a basis for the clinical diagnosis and carrier screening of SDS, we classified the pathogenicity of 64 different variants in 20,395 samples according to the 2015 ACMG guidelines. Four variants were found that could be classified as pathogenic or likely pathogenic. The c.258 +2T>C variant was detected most frequently (207 cases), followed by c.185_184TA>GT (33 cases), while the other two were null mutations that were only detected once (Tables 1 and S1). The total carrier frequency of *SBDS* pathogenic variants in the Chinese population was 1.19%. According to the Hardy-Weinberg equilibrium, the theoretical incidence of *SBDS*-related SDS in the Chinese population was 3.52 per 100,000, which was higher than that of other populations (0.5–1.5/100,000) [20]. This is the first study to analyze the carrier frequency of *SBDS* pathogenic variants in the Chinese population with large sample size. SDS is a severe early-onset disease, and the overall carrier frequency of *SBDS* gene pathogenic variants in the Chinese population is >1% [21]. Therefore, this study provided evidence for the *SBDS* gene as a candidate for screening potential carriers. Regarding the spectrum of pathogenic variants in the population, analyzing the common hotspot mutations, such as c.258+2T>C and c.185_184TA>GT, could effectively screen out most of the carriers.

Since PSVs could be informative for NGS read alignment for highly homologous genes, the more variants at *SBDS* PSVs loci occurred in a given NGS read, the more likely it was to be misaligned to *SBDSP1*, and the higher was the risk of false negatives. In the NGS data with 207 c.258+2T>C and 33 c.185_184TA>GT, it was found that most c.258+2T>C were detected in isolation (91.3%), and only a few detected *in-cis* with a variant at the c.201A PSV (8.7%). On the other hand, c.185_184TA>GT had a higher proportion (51.5%) detected *in-cis* with variants at the c.141C and/or c.201A PSVs, suggesting that NGS data reads containing c.185_184TA>GT have a greater chance of misalignment (Table 2). In our study, the allele balance of c.258+2T>C and c.185_184TA>GT both had a significant shift. Especially, nearly half of c.185_184TA>GT variants had allele balance < 0.3, the majority of which were arranged

**Table 1. Pathogenic or likely pathogenic variants detected *SBDS* gene in non-SDS group.**

| HGVS | Protein | Nᵃ | Frequency | Classification | Evidence |
|---|---|---|---|---|---|
| c.258+2T>C | p.? | 207 | 1.01% | Pathogenic | PVS1, PM3_Strong |
| c.183_184delTAinsCT | p.Lys62* | 33 | 0.16% | Pathogenic | PVS1, PS3, PM3_Strong |
| c.18dupC | p.Thr7Hisfs*43 | 1 | 0.00% | Likely Pathogenic | PVS1, PM2_Supporting |
| c.624+1G>A | p.? | 1 | 0.00% | Likely Pathogenic | PVS1, PM2_Supporting |

a The number of variant carriers.

**Table 2. The arrangement of variants detected at PVS loci of exon 2 region of *SBDS* gene.**

| 258+2T>C | 201A>G | 185_184TA>GT | c.141C>T | N[a] | Percentage |
|---|---|---|---|---|---|
| + | - | - | - | 199 | 96.14% |
| | + | - | - | 18 | 8.70% |
| - | - | + | - | 16 | 48.48% |
| - | + | | - | 11 | 33.33% |
| - | - | | + | 3 | 9.09% |
| - | + | | + | 3 | 9.09% |

a The number of variant carriers in non-SDS group.

*in-cis* with other variants at PSV loci (Fig 2). Allele balance bias made it difficult to distinguish whether the mutation reads were chimeric or misaligned. Furthermore, allele balance bias is an important quality control filter of NGS data. A pathogenic variant might be filtered if the allele balance is below the limit (usually 10%). Consequently, misdiagnosis and missed diagnosis were likely to occur, especially for individuals with atypical SDS symptoms. These two pathogenic variants were even excluded from The Exome Aggregation Consortium (ExAC) database due to failure to pass the variant quality score recalibration (VQSR) filter. It is worth noting that the common c.[185_184TA>GT; 258+2T>C] variant was not found in the Chinese population [22], which was inconsistent with the theoretically predicted carrier frequency. It was speculated that the *SBDS* NGS reads containing this variant were completely aligned to the *SBDSP1* locus due to the exchange of multiple PSVs, suggesting that the NGS analysis on c.[185_184TA>GT;258+ 2T>C] had limitations. Hence, it is necessary to establish a more effective analysis process to resolve the interference caused by the misalignment of NGS data in the analysis of *SBDS* variation.

In view of the above analysis of the *SBDS* variant and pathogenic variant spectrum in the Chinese population, we have found that the pathogenic variants of *SBDS* were mainly at functional PSV loci caused by short sequence exchange in the coding region and splice junction of exon two. Therefore, we formulated an NGS data processing procedure for *SBDS* in this particular region to overcome the NGS misalignment problem (Fig 3A). The primary strategy comprised the generation of a mathematical relationship between the *SBDS*:*SBDSP1* read-depth ratio (RR) of each PSV locus and the total *SBDS*+*SBDSP1* read-depth (TR) of each PSV locus. As shown in Fig 3, with wild-type *SBDS* and *SBDSP1*, NGS reads could be accurately aligned

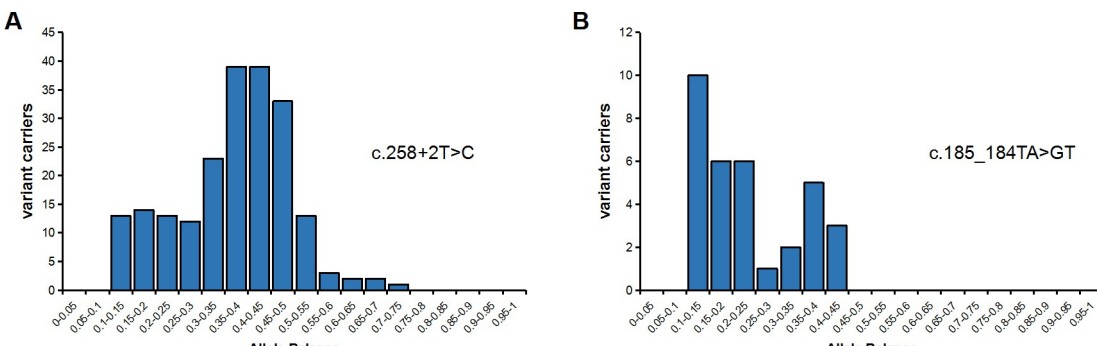

**Fig 2. Allele balance of *SBDS* gene pathogenic variants c.258+2T>C (n = 207) and c.185_184TA>GT (n = 33) in the non-SDS group.** Note the y-axis for the number of heterozygous variants carriers. Variants with low allele balance (<0.3) are considered to be having misalignment issue.

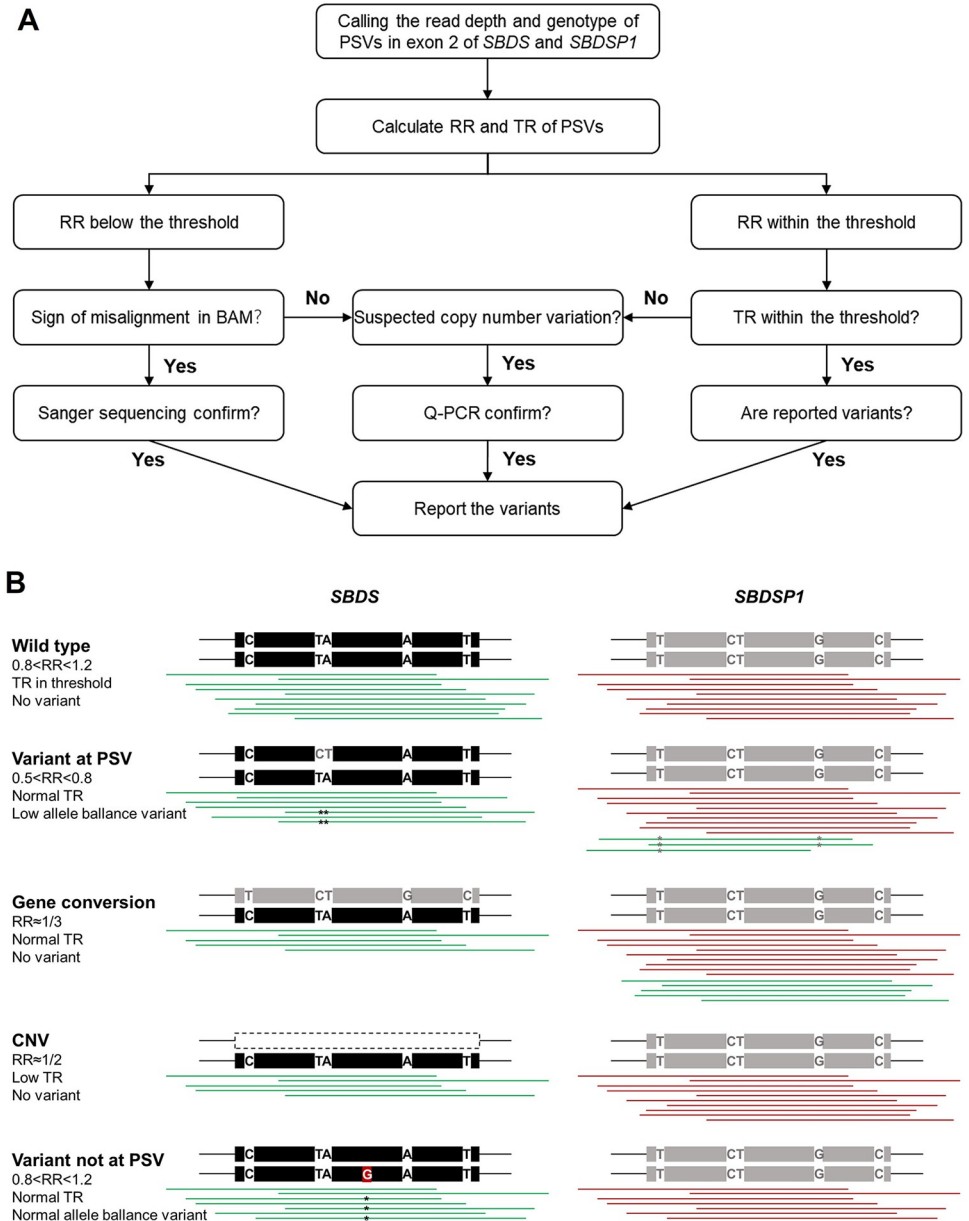

**Fig 3. Interpretation of NGS data of exon two of *SBDS* gene. A.** The analytical decision tree for detecting variants at exon two of the *SBDS* gene. The RR and TR values at PSVs loci were first calculated. If the RR values are low, BAM files should be checked for signs of misalignment, which should be confirmed with Sanger sequencing; If the RR and TR values are normal, any reportable variants can be reported directly; If the TR and RR values indicate a copy number variation, quantitative PCR is required to confirm. **B.** Overview of misalignment caused by variants at the coding region and splice junction of exon two of *SBDS* gene. There are five nucleotide differences between *SBDS* and *SBDSP1*. When one or multiple variants occur at PSV loci, alignment programs map the reads to the wrong position. From top to bottom, wild-type *SBDS* gene with correct mapping result; *SBDS* gene with a heterozygous variant (c.185_184TA>GT) at PSV loci; *SBDS* gene with multiple *in-cis* heterozygous variants at PSV loci; *SBDS* gene with large heterozygous deletion; *SBDS* gene with variant at non-PSV loci.

to the correct locus, and the RR and TR values of all PSV loci were averagely 1 (between 0.8–1.2) and 4, respectively. When the gene conversion occurred at one or multiple PSV loci, the RR values of PSV loci decreased substantially to lower than 1. Under these circumstances,

misalignment could be checked based on BAM files. For example, one or multiple variants with a low allele balance at PSV loci appeared in *SBDS* reads, and a portion of *SBDS* reads was misaligned to *SBDSP1*. The more PSV loci involved in this micro-conversion event, the higher chances of *SBDS* reads being misaligned to *SBDSP1*, and even no mutations could be identified in *SBDS* or *SBDSP1*. At this time, the RR values of multiple PSV loci could decrease significantly (as low as 1/3), and the TR value of PSVs loci was not significantly different from that of other WES data in the same capture pool. Then, a Sanger sequencing was required to detect this suspected micro-conversion event. For individuals carrying a heterozygous deletion of the *SBDS* gene, the RR values of multiple PSV loci were also reduced (~1/2), but the TR value was lower than that of other WES data in the same capture pool, which was different from the misalignment. Then, a quantitative PCR was required to detect this suspected copy number variation. If a reportable variant was found in exon two of the *SBDS* gene but at non-PSV position and there was no sign of misalignment or copy number variation based on the RR and TR values of all PSV loci, the variant could be reported directly, needing no further confirmation. The novel strategy had the potential to find misalignment events in *SBDS* NGS analyses and provides an additional opportunity to manually check NGS data and apply orthogonal methods.

Using the above *SBDS* NGS analysis strategy, ten homozygous or compound heterozygous pathogenic variants in the *SBDS* gene were detected from WES data of 147 cases in the suspected SDS group. Among them, c.258+2T>C had the highest number of detections (10 cases/100%), followed by c.185_184TA>GT (5 cases per 50%), c.[185_184TA>GT;258+2T>C] (1 case/10%) and c.120del (1 case/10%); three homozygous pathogenic variants were all c.258+2T>C. Our analysis strategy improved the sequencing results of 40% (4/10) of the samples.

In one case (case 1), a c.258+2T>C variant in homozygous form was initially identified in the general NGS analysis. Further comparison with other WES data in the same capture pool revealed reduced RR values (approximately 1/3) and normal TR values were found at multiple PSV loci across exon two of *SBDS* gene, which indicated a potential misalignment event in the NGS analysis (Figs 4 and 5). Post-verification by Sanger sequencing, the case was ultimately proved to be a compound heterozygote of c.258+2T>C and c.[141C>T;185_184TA>GT;201A>G;258+2T>C] (Fig 5). The reason for the false homozygosity result was that NGS reads containing *SBDS* variants c.[141C>T;185_184TA>GT;201A>G;258+2T>C] were all misaligned to the *SBDSP1* locus.

In two cases, a c.258+2T>C variant in heterozygous form was initially identified in general NGS analysis. Further comparison with other WES data in the same capture pool revealed reduced RR values, and normal TR values were found at multiple PSV loci in exon two of *SBDS* gene, which indicated a potential misalignment event in the NGS analysis. For example, in one representative case (case 2), the c.185_184TA>GT and c.201A>G variants were found in a few reads (3/35) at the *SBDS* PSV loci when we further manually checked the BAM file, which was not identified because they were filtered out by the mapping quality threshold (10%). At the same time, the n.489T>C variant was found at the *SBDSP1* PSV locus, which might have been misaligned from the NGS reads of *SBDS* (Figs 6 and 7). Post-verification by Sanger sequencing, the case was ultimately proved to be a compound heterozygote of c.258+2T>C and c.[185_184TA>GT; c.201A>G] (Fig 7). Thus, our strategy for *SBDS* NGS analyses provided an additional opportunity to manually check NGS data, which could be helpful for finding and distinguishing the authenticity of variants with extreme allele balances.

Due to the misalignment of NGS data, *SBDS* variants sometimes have abnormal allele balances, which would then manifest as unexpected segregation patterns or be misunderstood as somatic mosaicism. However, abnormal genetic events like these are really rare in reported cases. This was also supported by the fact that all *SBDS* variants from the suspected SDS group

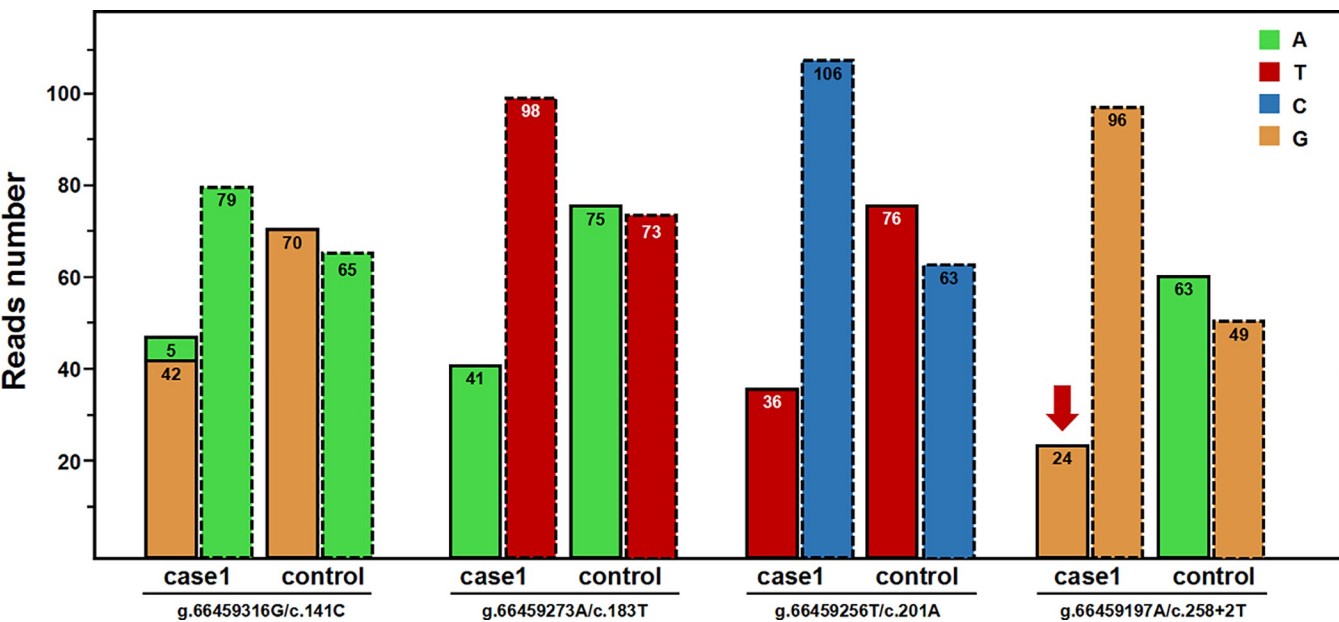

**Fig 4. Read-depths of the four genomic positions at the PSV loci of *SBDS* gene and the corresponding loci of *SBDSP1* gene (case 1).** Bars with a dashed outline represent the read-depths at the loci of *SBDS* gene; the adjacent bars with a dotted outline represent the read-depths at the corresponding loci of *SBDSP1* gene. Control represents the mean read-depth of other WES data in the same capture pool.

were inherited from heterozygous parents, proved by orthogonal methods. In particular, we found that one case (case 3) had a homozygous c.258+2T>C variant only inherited from a heterozygous father. The WES data showed normal RR and TR values at the *SBDS* PSV loci, which ruled out the possibility of large fragment DNA deletion or reads misalignment in the *SBDS* gene (Figs 8 and 9). Further, a multiplex STR testing was done to explain this unexpected

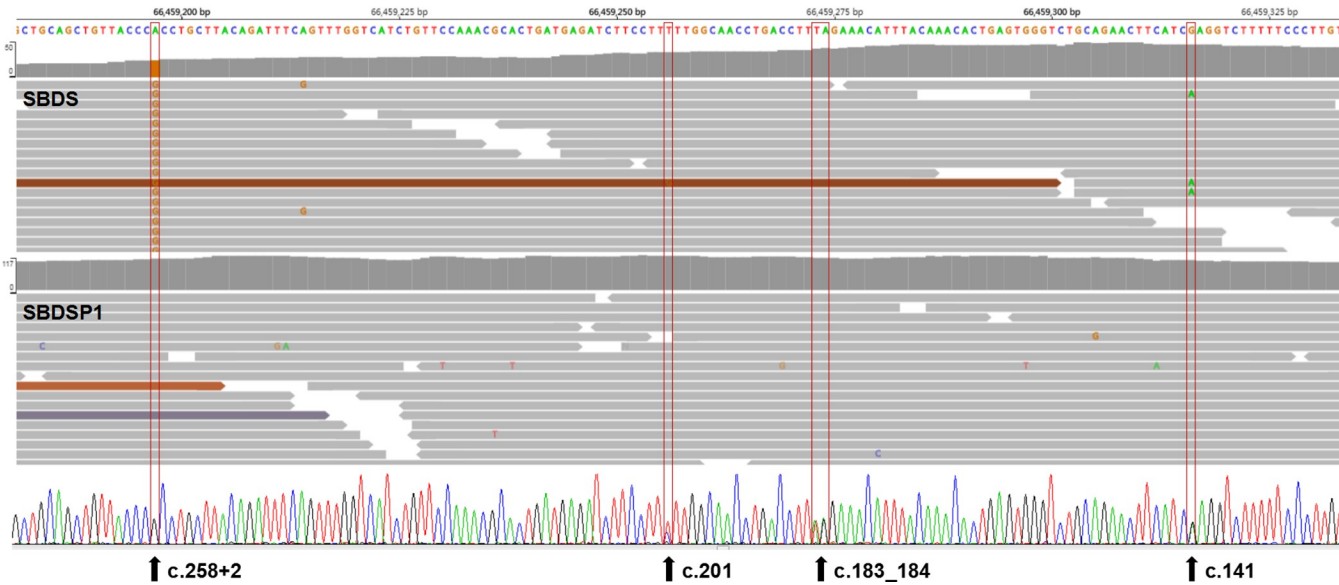

**Fig 5. Aligned NGS reads from WES data (top) and Sanger sequencing peak map (bottom) of case 1.** NGS reads are shown by the IGV software using genomic coordinates. The WES data showed that case 1 was homozygous for the 258+2T>C variant. The read-depth of exon 2 region of *SBDS* gene is reduced, and the read-depth of the corresponding region of *SBDSP1* gene is increased. Sanger sequencing results indicated that case 1 was a compound heterozygote for c.258+2T>C and c.[141C>T;183_184TA>GT;201A>G;258+2T>C].

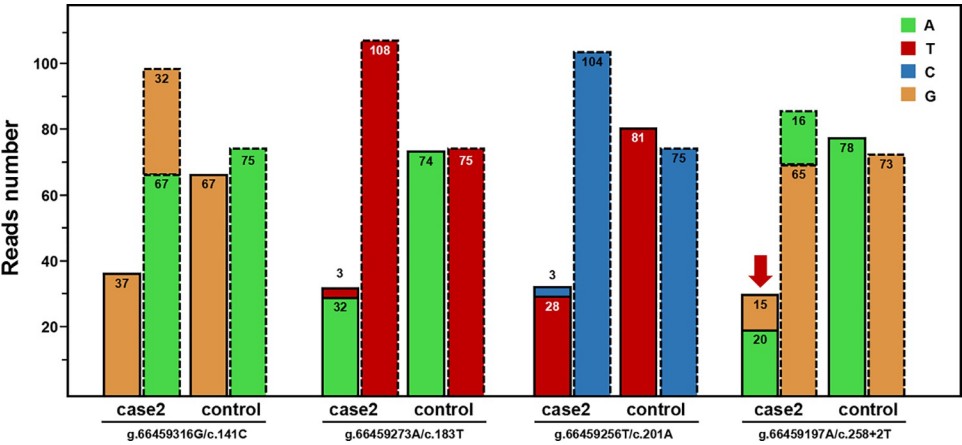

**Fig 6. Read-depths of the four genomic positions at the PSV loci of *SBDS* gene and the corresponding loci of *SBDSP1* gene (case 2).** The bars with a dashed outline represent the read-depths at the loci of *SBDS* gene; the adjacent bars with a dotted outline represent the read-depths at the corresponding loci of *SBDSP1* gene. Control represents the mean read-depth of other WES data in the same capture pool.

pattern of segregation, and the results showed that the homozygous c.258+2T>C variant was caused by the paternal uniparental disomy of chromosome seven (Fig 10). This is the first report of uniparental disomy caused a homozygous pathogenic variant in SDBS. It is suggested that our strategy for *SBDS* NGS analyses provided an additional opportunity to apply more appropriate orthogonal methods.

## 4. Conclusion

Our proposed NGS variant data analysis method, based on the read-depth analysis of the PSV loci of *SBDS* and its pseudogene, could effectively overcome the misalignment problem in

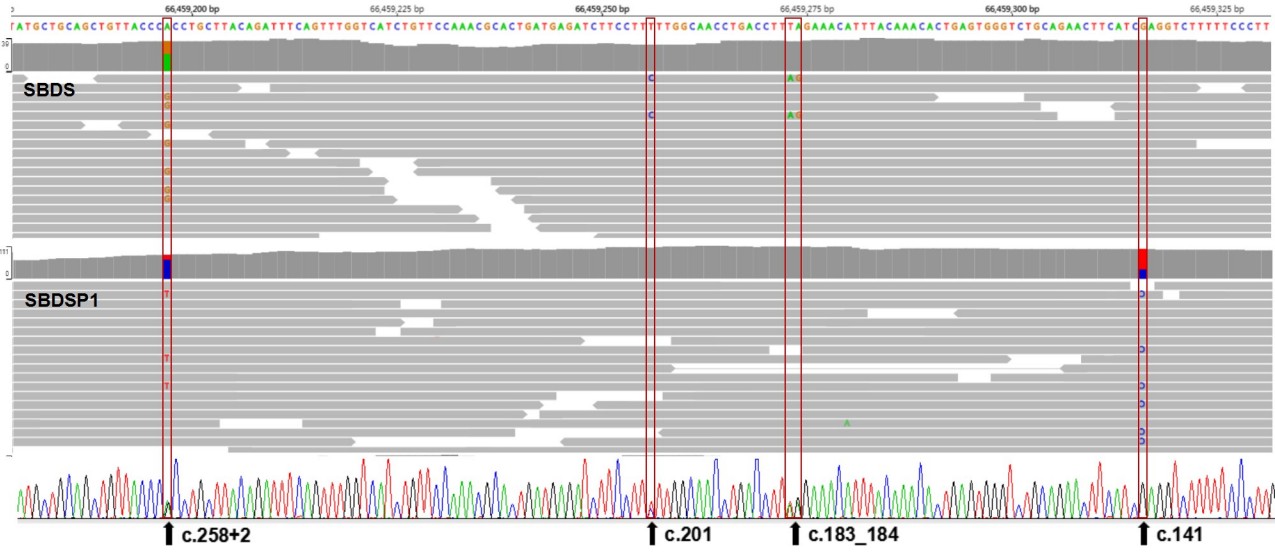

**Fig 7. Aligned NGS reads from WES data (top) and Sanger sequencing peak map (bottom) of case 2.** NGS reads are shown by the IGV software using genomic coordinates. The WES data showed that case 2 was homozygous for the 258+2T>C variant, and reads with 201A>G and 183_184TA>CT were disregarded as low-quality reads. The read-depth of the exon 2 region of *SBDS* gene is reduced and the read-depth of the corresponding region of *SBDSP1* gene is increased. Sanger sequencing results indicated that case 2 was a compound heterozygote for c.258+2T>C and c.[183_184TA>GT;201A>G].

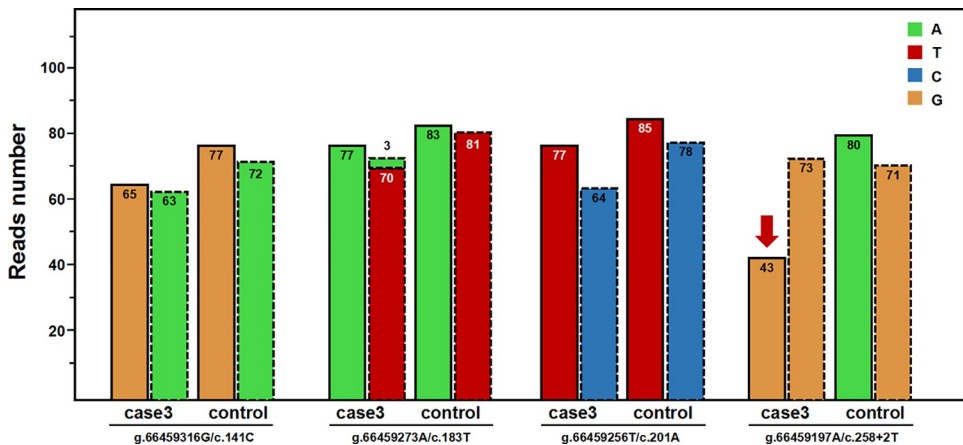

**Fig 8. Read-depths of the four genomic positions at the PSV loci of *SBDS* gene and the corresponding loci of *SBDSP1* gene (case 3).** The bars with a dashed outline represent the read-depths at the loci of *SBDS* gene; the adjacent bars with a dotted outline represent the read-depths at the corresponding loci of *SBDSP1* gene. Control represents the mean read-depth of other WES data in the same capture pool.

*SBDS* NGS data analysis and improve detection rates of *SBDS* pathogenic variants. Based on the Chinese mutation spectrum of *SBDS*, the novel method developed in this study not only focused on the most common variants in the *SBDS* gene exon 2 but could also be used to detect structural variation (SV) in *SBDS*. Furthermore, the method was useful for performing more efficient molecular diagnosis, the variant spectrum study, and large-scale carrier screening of SDS. In addition, *SBDS* gene Exon 2 is only a representative of regional misalignment, and such micro-conversion events may also occur between genes and their pseudogenes, especially within some large exons. For example, micro-conversion involved part of the E6 cluster of the *CYP21A2* gene, exon 28 of *VWF* (1379bp in length, 97.1% in homology), and exon 9 of *ARHGAP21* (1896 bp in length, 98.6% in homology). In this circumstance, exon-level misalignment analysis may not reach a statistical threshold but present negative or ambiguous results. Our NGS variant data analysis method can override such constraints.

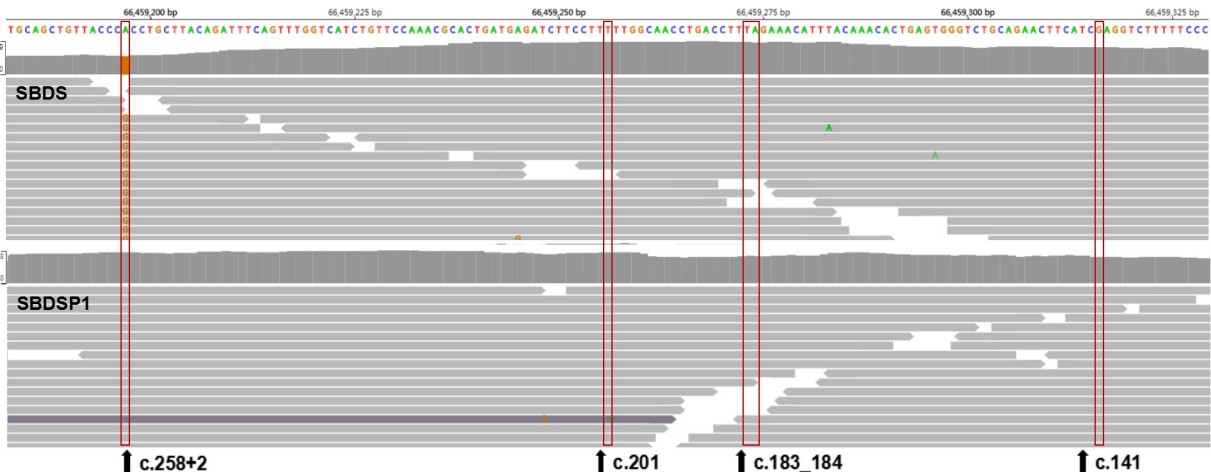

**Fig 9. Aligned NGS reads from WES data of case 3.** NGS reads are shown by the IGV software using genomic coordinates. The WES data showed that case 3 was homozygous for the 258+2T>C variant. The read-depth of the exon 2 region of *SBDS* gene was not significantly different from that of the corresponding region of *SBDSP1* gene.

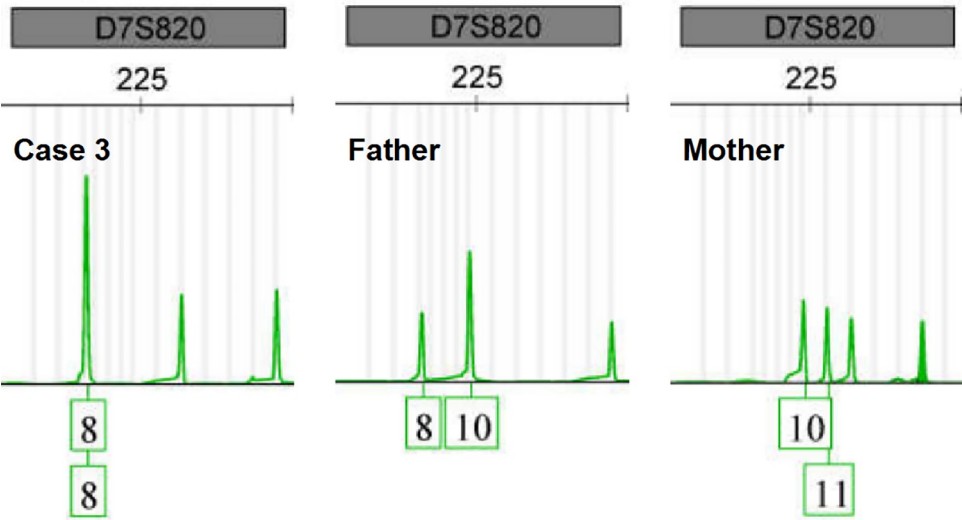

**Fig 10. STR analysis result of chromosome 7 (case 3 family).** STR analysis showed that all STR polymorphisms of case 3 are inherited from both parents, expect the STR polymorphisms of chromosome 7, which are proved to be paternally inherited.

## Supporting information

**S1 Table. The SBDS mutation spectrum from WES data of 20,395 samples from the Chinese population.**
(XLSX)

**S2 Table. Haplotype counts of c.141C>T of SBDS gene and its corresponding SBDSP1 gene variant in non-SDS group (N = 20395).**
(DOCX)

## Author Contributions

**Data curation:** Li Zhang, Yuzhen Qiang, Kaiyu Wang.

**Formal analysis:** Kaiyu Wang.

**Investigation:** Kaiyu Wang.

**Methodology:** Kaiyu Wang.

**Resources:** Dong Wu.

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
