## [Decision Letter · Decision Letter 0]

22 Apr 2022

PONE-D-21-31295

Improved detection of SBDS gene mutation by a new method of next-generation sequencing analysis based on the Chinese mutation spectrum

PLOS ONE

Dear Dr. Wang,

Thank you for submitting your manuscript to PLOS ONE. After careful consideration, we feel that it has merit but does not fully meet PLOS ONE’s publication criteria as it currently stands. Therefore, we invite you to submit a revised version of the manuscript that addresses the points raised during the review process.

We look forward to receiving your revised manuscript.

Kind regards,

Muhammad Abdul Rehman Rashid, PhD

Academic Editor

PLOS ONE

Reviewers' comments:

Reviewer's Responses to Questions

**Comments to the Author**

1. Is the manuscript technically sound, and do the data support the conclusions?

Reviewer #1: Yes

Reviewer #2: Partly

2. Has the statistical analysis been performed appropriately and rigorously? 

Reviewer #1: Yes

Reviewer #2: I Don't Know

3. Have the authors made all data underlying the findings in their manuscript fully available?

Reviewer #1: Yes

Reviewer #2: No

4. Is the manuscript presented in an intelligible fashion and written in standard English?

Reviewer #1: Yes

Reviewer #2: Yes

5. Review Comments to the Author

Reviewer #1: The NGS variation data analysis method based on the reading depth analysis of SBDS and its pseudogene PSV loci proposed in this paper can effectively overcome the problem of dislocation in SBDSNGS data analysis and improve the detection rate of SBDS pathogenic variation. This method is based on the mutation spectrum of Chinese SBDS, and has certain clinical significance for more effective molecular diagnosis, mutation spectrum study and large-scale carrier screening of SDS.

Reviewer #2: Dear Authors,

Overall the manuscript is interesting and valuable to assist NGS data analysis. However, my major criticize is the fact that authors claim a creation of a new method and a poor description of it is made. Thus making almost impossible to replicate it. Authors should include more details of the new method created. Further, a discussion regarding the applicability of this method to other diseases or populations is also important.

Other specific comments are refered in the pdf file attached

6. PLOS authors have the option to publish the peer review history of their article (what does this mean?). If published, this will include your full peer review and any attached files.

Reviewer #1: No

Reviewer #2: No

---

## [Author Response · Author response to Decision Letter 0]

12 May 2022

Dear Editors and Reviewers,

Thank you for your letter and the reviewers’ comments concerning our manuscript entitled “Improved detection of SBDS gene mutation by a new method of next-generation sequencing analysis based on the Chinese mutation spectrum” (PONE-D-21-31295). Those comments are valuable and very helpful. We have read through comments carefully and have made corrections. Based on the instructions provided in your letter, we uploaded the file of the revised manuscript. Accordingly, we have uploaded a copy of the original manuscript with all the changes highlighted by using the track changes mode in MS Word. The main corrections in the paper and the responds to the reviewer’s comments are as flowing:

Response to reviewer #1:

“The NGS variation data analysis method based on the reading depth analysis of SBDS and its pseudogene PSV loci proposed in this paper can effectively overcome the problem of dislocation in SBDSNGS data analysis and improve the detection rate of SBDS pathogenic variation. This method is based on the mutation spectrum of Chinese SBDS, and has certain clinical significance for more effective molecular diagnosis, mutation spectrum study and large-scale carrier screening of SDS.”

Response: Thank you for your summary. We really appreciate your efforts in reviewing our manuscript.

Response to reviewer #2:

“Overall the manuscript is interesting and valuable to assist NGS data analysis. However, my major criticize is the fact that authors claim a creation of a new method and a poor description of it is made. Thus making almost impossible to replicate it. Authors should include more details of the new method created. Further, a discussion regarding the applicability of this method to other diseases or populations is also important.”

Response: Thank you for your summary. We really appreciate your efforts in reviewing our manuscript. We have revised the manuscript accordingly. Our point-by-point responses are detailed below.

Q1: Authors should include more details of the new method created.

Response: We apologize for not describing the method clearer. we have added a more detailed interpretation and a flow diagram (Please see Figure3) to describe the procedure of our strategy.

Q2: A discussion regarding the applicability of this method to other diseases or populations is also important.

Response: Thank you for the suggestion. As suggested by reviewer, we have added the suggested content to the manuscript. Please see “4 Conclusion”

Q3: “How this is done? ”– the comment on “The average read-depths of each exon of SBDS and the corresponding region of SBDSP1 were extracted to estimate the frequency of conversion between SBDS and SBDSP1.”

Response: Thank you for the suggestion. We have added the information required. Please see “Whole-Exome Sequencing” section in “2 Materials and Methods”.

Q4: “How is the processing procedure?”; “where is the mathematical model?”; “So, the mutations previously referred were from which gene?”; “so, this were not mutations? I'm confused”—the comments on “SBDS NGS analysis strategy”.

Response: We apologize for not describing the strategy clearer. we have added a more detailed interpretation and a flow diagram (Please see Figure3) to describe the procedure of our strategy.

Sincerely.

Kaiyu Wang

---

## [Editor Report · Decision Letter 1]

13 May 2022

Improved detection of SBDS gene mutation by a new method of next-generation sequencing analysis based on the Chinese mutation spectrum

PONE-D-21-31295R1

Dear Dr. Wang,

We’re pleased to inform you that your manuscript has been judged scientifically suitable for publication and will be formally accepted for publication once it meets all outstanding technical requirements.

Kind regards,

Muhammad Abdul Rehman Rashid, PhD

Academic Editor

PLOS ONE
---

## [Editor Report · Acceptance letter]

4 Aug 2022

PONE-D-21-31295R1 

Improved detection of *SBDS* gene mutation by a new method of next-generation sequencing analysis based on the Chinese mutation spectrum 

Dear Dr. Wang:

I'm pleased to inform you that your manuscript has been deemed suitable for publication in PLOS ONE. Congratulations! Your manuscript is now with our production department. 

Kind regards, 

on behalf of

Dr. Muhammad Abdul Rehman Rashid 

Academic Editor

PLOS ONE